# Estrogenic Effects of Extracts and Isolated Compounds from Belowground and Aerial Parts of *Spartina anglica*

**DOI:** 10.3390/md19040210

**Published:** 2021-04-11

**Authors:** Sullim Lee, Geum Jin Kim, Hyukbean Kwon, Joo-Won Nam, Ji Yun Baek, Sang Hee Shim, Hyukjae Choi, Ki Sung Kang

**Affiliations:** 1Department of Life Science, College of Bio-Nano Technology, Gachon University, Seongnam 13120, Korea; sullimlee@gachon.ac.kr; 2College of Pharmacy, Yeungnam University, Gyeongsan 38541, Korea; canta87@ynu.ac.kr (G.J.K.); zero9602@gmail.com (H.K.); jwnam@yu.ac.kr (J.-W.N.); 3Research Institute of Cell Culture, Yeungnam University, Gyeongsan 38541, Korea; 4College of Korean Medicine, Gachon University, Seongnam 13120, Korea; wldbsttn@naver.com; 5Department of Food Science, Gyeongnam National University of Science and Technology, Jinju 52725, Korea; 6Natural Products Research Institute, College of Pharmacy, Seoul National University, 1 Gwanak-ro, Gwanak-gu, Seoul 08826, Korea; sanghee_shim@snu.ac.kr

**Keywords:** estrogenic effects, *Spartina anglica*, 1,3-Di-*O*-*trans*-feruloyl-(-)-quinic acid, MCF-7

## Abstract

Menopause, caused by decreases in estrogen production, results in symptoms such as facial flushing, vaginal atrophy, and osteoporosis. Although hormone replacement therapy is utilized to treat menopausal symptoms, it is associated with a risk of breast cancer development. We aimed to evaluate the estrogenic activities of *Spartina anglica* (SA) and its compounds and identify potential candidates for the treatment of estrogen reduction without the risk of breast cancer. We evaluated the estrogenic and anti-proliferative effects of extracts of SA and its compounds in MCF-7 breast cancer cells. We performed an uterotrophic assay using an immature female rat model. Among extracts of SA, belowground part (SA-bg-E50) had potent estrogenic activity. In the immature female rat model, the administration of SA-bg-E50 increased uterine weight compared with that in the normal group. Among the compounds isolated from SA, 1,3-di-*O*-*trans*-feruloyl-(-)-quinic acid (**1**) had significant estrogenic activity and induced phosphorylation at serine residues of estrogen receptor (ER)α. All extracts and compounds from SA did not increase MCF-7 cell proliferation. Compound **1** is expected to act as an ERα ligand and have estrogenic effects, without side effects, such as breast cancer development.

## 1. Introduction

Female hormones are produced by mature follicles in the ovary, promote the development of secondary sexual characteristics in women, and play a role in thickening the membrane for the implantation of the fertilized egg in the uterine wall [1,2]. Owing to the action of female hormones, the uterine wall collapses and thickens periodically, which is called menstruation [3]. The ovaries genetically regulate the fertility of women by the generation of female hormones; however, in menopause, this function is lost, which leads to various physiological changes [4,5]. Recently, the number of women affected by menopause (the period after menstruation ends) is increasing in line with an increase in lifespan. Reportedly, women now spend one-third of their lives in menopause [6]. Menopause presents with symptoms such as facial flushing, vaginal atrophy, vaginal dryness, insomnia, and osteoporosis owing to decreased estrogen production from the ovaries. It can lead to a lower quality of life and depression [7,8,9,10]. 

Among the female hormones, estrogen, which plays an important role in the regulation of the female menstrual cycle, regulates the transcription of target genes by binding to the estrogen receptor (ER) [11,12]. Menopause is caused by decreases in estrogen production [13]. Hormone replacement therapy (HRT), which is the administration of female hormones to counter the lack of estrogen in the body in menopausal women, is used to treat menopausal symptoms [14]. However, HRT has been reported to have side effects that increase the risk of breast cancer development [15,16]. Therefore, research on treatments that can reduce the risk of breast cancer [17] and alleviate menopausal symptoms is required.

Phytochemicals that can act as potential ER agonists or antagonists are called phytoestrogens [18]. Phytoestrogens often serve as selective estrogen receptor modulators and, therefore, have organ-specific estrogen activity [19]. They exhibit antagonistic activity in the uterus and breast and are active in the bones, cardiovascular system, and brain [20]. Additionally, plant-derived chemicals containing these compounds have potential for HRT because of their ER-mediated molecular mechanisms of action [21]. Phytoestrogens are secondary metabolites biosynthesized in plants that contain at least one hydroxyl group. The interactions of these hydroxyl groups with the ligand-binding domains of ERα and ERβ lead to receptor activation [22]. Isoflavones, stilbenes, and lignans have been reported to be phytoestrogens, and they have been demonstrated to relieve various symptoms of menopause [23,24,25].

*Spartina anglica* C. E. Hubb (Poaceae) is a perennial halophyte in the coastal areas of England and France [26]. This herbaceous plant, called cord-grass, has dense roots and has been used for protection against coastal erosion [27]. However, it was recently reported to disturb the mudflat ecosystem [28]. Since the first report on the occurrence of *S. anglica* in the southern parts of Ganghwa Island, South Korea, in 2012, its range has expanded rapidly and the Korean government has designated this invasive halophyte as an “unintroduced species” [29]. Owing to the rapidly expanding range of this plant, the cost of waste biomass treatment is increasing. To decrease the management cost, we attempted to determine the bioactivity of this plant and its constituents. Recently, we reported the antioxidant, tyrosinase inhibitory, and pancreatic lipase inhibitory activities of the extract and phytochemicals of *S. anglica* [30]. However, their estrogenic activities have not been studied. 

Thus, we investigated the estrogenic effects of the extracts and fractions of *S. anglica* in the MCF-7 human breast cancer cell line. Isolation steps were conducted to identify potential candidates for the treatment of estrogen reduction and obtain five secondary metabolites from *S. anglica* (Figure 1). We also evaluated the effects of these active compounds against uterine hypertrophy in the immature female rat model. In the present study, we describe the estrogenic effects of compounds from *S. anglica* in MCF-7 cells and an immature female rat model. 

## 2. Results and Discussion 

### 2.1. Estrogenic Effects of Different Extracts from Belowground and Aerial Parts of S. anglica on MCF-7 Cells

To evaluate the potential estrogenic activity of the extracts of *S. anglica*, Soto’s E-screen assay using an ER-positive MCF-7 human breast cancer cell line was performed [31]. Queens One tablet (tab.), which was used as a positive control, increased MCF-7 cell proliferation in a concentration-dependent manner (25 μg/mL: 115.2 ± 6.3%, 50 μg/mL: 129.0 ± 4.6%, 100 μg/mL: 132.5 ± 5.9%; Figure 2). Moreover, because proliferation was significantly suppressed in the presence of ICI 182,780, which is an ER antagonist, the proliferation caused by Queens One Table was regarded as estrogen-responsive proliferation. 

Among the tested extracts, the 50% ethanol extract from the belowground part of *S. anglica* (SA-bg-E50) induced cell proliferation at 25–100 μg/mL (25 μg/mL: 118.4 ± 3.4%, 50 μg/mL: 110.0 ± 2.8%, 100 μg/mL: 108.4 ± 4.5%); however, the effect was not concentration-dependent. It seems that SA-bg-E50 contains the estrogenic phytochemicals along with their competitive inhibitors. These two opposite actions can cause concentration independent estrogenic activities of SA-bg-E50. The proliferation in the groups treated with 25 and 50 μg/mL was significantly suppressed to 95.5 ± 4.1% and 93.2 ± 3.9%, respectively, by ICI 182,780, whereas the 100 μg/mL-treated group did not show a significance. These results indicate that SA-bg-E50 has significant estrogenic activity at concentrations of 25–100 μg/mL. The 30% ethanol extract from the aerial part of *S. anglica* (SA-a-E30) increased cell proliferation in a concentration-dependent manner (25 μg/mL: 111.4 ± 4.6%, 50 μg/mL: 112.2 ± 8.3%, 100 μg/mL: 115.7 ± 8.5%). Moreover, proliferation was partially suppressed in the presence of ICI 182,780 (25 μg/mL: 94.3 ± 5.5%, 50 μg/mL: 96.9 ± 5.5%, 100 μg/mL: 107.7 ± 10.0%). These results suggest that SA-a-E30 does not have meaningful estrogenic activity, and it is considered a partial agonist of the ER. Taken together, SA-bg-E50 has potent estrogenic activity compared with the other extracts tested in this study; thus, we focused on SA-bg-E50 in subsequent experiments.

### 2.2. Uterotrophic Activity of 50% Ethanol Extract from Belowground Part of S. anglica in the Immature Rat

To determine the potential of SA-bg-E50 to have estrogenic effects, we evaluated the estrogenic effect of SA-bg-E50 in an immature female rat model. 17β-Estradiol (10 μg/kg) was subcutaneously injected for 3 days in the control group, and SA-bg-E50 (200 mg/kg) was orally administered for 3 days in the SA-bg-E50 group. 

The uterine weight of the positive control group was significantly increased, to 50.3 ± 3.1 mg, compared with that of the normal group (41.5 ± 1.6 mg). After the administration of SA-bg-E50, the uterine weight was 49.0 ± 2.5 mg. The results indicate that SA-bg-E50 hypertrophies the uterus owing to its estrogenic activity (Figure 3). 

### 2.3. Structure Elucidation of Compounds ***1–5*** from S. anglica

Compound **1** exhibited a deprotonated [M − H]^−^ ion at *m/z* 543.2 in low-resolution electron spray ionization mass spectrometry (LR-ESI-MS). Based on the interpretation of the 1D and 2D nuclear magnetic resonance (NMR) data of **1** (Appendix A), two sets of vinyl groups, two sets of 1,3,4-trisubstituted benzene rings, and a quinic acid moiety were identified. The ^1^H-^1^H coupling constants of H-7′/H-8′ (7.53 (1H, d, *J* = 15.9 Hz, H-7′)/6.26 (1H, d, *J* = 15.9 Hz, H-8′)) and H-7″/H-8″ (7.54 (1H, d, *J* = 15.9 Hz, H-7″)/6.16 (1H, d, *J* = 15.9 Hz, H-8″)) were the *trans* configurations of two sets of olefinic protons. The proton signals for the two aromatic systems with 1,3,4-trisubstitution were observed at δ_H_ 6.93 (1H, d, *J* = 1.9, H-2′), 6.88 (1H, dd, *J* = 8.2, 1.9, H-6″), 6.83 (1H, d, *J* = 1.9, H-2″), 6.75 (1H, dd, *J* = 8.2, 1.9, H-6′), 6.67 (1H, d, *J* = 8.2, H-5″), and 6.57 (1H, d, *J* = 8.1, H-5′). The proton and carbon resonances at δ_H/_δ_C_ 3.67 (3H, s)/56.18 ppm and 3.60 (3H, s)/56.08 ppm indicated the presence of two methoxy groups. Through the analysis of heteronuclear multiple bond correlation (HMBC), two ferulate units were identified. The ^1^H and ^13^C NMR spectra of **1** showed the characteristic resonances of a quinic acid at δ_H/_δ_C_ 5.36/73.21, 4.28/67.80, 3.64/75.31, 2.97/2.31/32.74, and 2.54/1.84/41.46 ppm. One feruloyl residue was connected to the C-1 of quinic acid, as shown by the downfield shift in ^13^C NMR at δ_C_ 81.12 (C-1) ppm. The position of the other ferulate unit was determined by HMBC from H-3 to C-9″ (Figure 4). The numbering system of acyl-quinic acid was critical because of the confusing nomenclature caused by mixing IUPAC with non-IUPAC numbering [32]. A number of reported publications related to acyl-quinic acid were identified with inconsistencies in numbering and/or structure with IUPAC numbering [33,34]. The planar structure of **1** was determined to be 1,3-di-*O*-*trans*-feruloyl quinic acid by the IUPAC nomenclature of the quinic acid moiety. The two of vicinal coupling constants of H-5 were observed to be large (^3^*J*_H-4,H-5_ = 9.6 Hz, ^3^*J*_H-5,H-6ax_ = 11.3 Hz) which supported the three protons of H-4, H-5 and H-6_ax_ as axial position (Figure 4). In addition, other coupling constants, including long-range coupling through a 1,3-diequatorial interaction (W-coupling) between H-2_eq_ and H-6_eq_, were well-matched with those of quinic acid, rather than with those of *epi*-quinic acid (Figure 4) [35]. The comparison of the specific rotation of **1** with that of similar chemical compounds (cynarine; 1,3-dicaffeoylquinic acid) supported the absolute configuration of quinic acid in **1** as (−)-quinic acid (experimental; [α]$\begin{array}{c} 25\\ D \end{array}$ = −9.7 (*c* = 0.1, MeOH), literature; [α]$\begin{array}{c} 25\\ D \end{array}$ = −59.0 (*c* = 4, MeOH) [36]. Compound **1** was identified as 1,3-di-*O*-*trans*-feruloyl-(−)-quinic acid with the absolute configuration of 1*R*, 3*R*, 4*S*, 5*R* (1α, 3α, 4α, 5β).

Compound **2** exhibited a protonated [M + H]^+^ ion at *m/z* 433.4 in the LR-ESI-MS spectrum. In the ^1^H NMR spectrum, compound **2** showed the characteristic features of flavonoid glycoside; a singlet resonance at δ_H_ 6.42 ppm (H-8, ring A), two *ortho* coupled signals [δ_H_ 7.90 (d, *J* = 8.8 Hz, H-2′/6′) and 6.91 ppm (d, *J* = 8.8 Hz, H-3′/5′)] of a 1,4-substituted phenyl group (ring B), a singlet proton signal at δ_H_ 6.71 ppm (H-3, ring C), an anomeric proton signal [4.58 ppm (d, *J* = 9.8 Hz, H-1″)], and protons attached to oxygenated carbons (δ_H_ 4.10–3.00 ppm) of sugar units. The sugar unit was identified to be C-β-D-glucopyranose based on the oxygenated carbons at δ_C_ 81.51 (C-5″), 79.05 (C-3″), 73.27 (C-1″), 70.66 (C-2″), 70.26 (C-4″), and 61.48 (C-6″) ppm and the large coupling constant value between H-1″ and H-2″. The carbon resonances at δ_C_ 109.10 (C-6) and 94.00 (C-8) ppm supported 6-C-glycoside rather than 8-C-glycoside. The comparison of experimental and reported [37] spectroscopic data, including MS and 1D NMR spectra, led to the identification of **2** as isovitexin (Appendix A).

Compounds **3**–**5** were identified as *p*-hydroxybenzaldehyde (**3**), *N*-*trans*-feruloyltyramine (**4**), and *p*-coumaric acid (**5**) by comparing 1D NMR and LR-ESI-MS data with those reported in the literature (Appendix A) [38,39,40].

### 2.4. Estrogenic Effects of Compounds Isolated from Belowground Part of S. anglica on MCF-7 Cells

We evaluated the estrogenic potential of the compounds isolated from belowground part of *S. anglica* in MCF-7 cells. 1,3-Di-*O*-*trans*-feruloyl-(-)-quinic acid (**1**) induced significant cell proliferation in a concentration-dependent manner (50 μM: 109.0 ± 4.7%, 100 μM: 115.4 ± 2.1%). Cell proliferation was significantly suppressed to 94.1 ± 2.6% (50 μM) and 95.9 ± 6.8% (100 μM) by ICI 182,780. These results indicate that compound **1** has significant estrogenic activity at concentrations of 50 and 100 μM. Isovitexin (**2**) also increased cell proliferation in a concentration-dependent manner (50 μM: 106.9 ± 4.1%, 100 μM: 111.2 ± 1.9%). Cell proliferation was suppressed to 89.8 ± 9.9% (50 μM) and 93.4 ± 7.1% (100 μM) by ICI 182,780, however it was not significant. (Figure 5). Taken together, compound **1** has potent significant estrogenic activity compared with other compounds; therefore, we focused on the bioactivity of compound **1** in subsequent investigations. 

ERα, an estrogen receptor 1, is a ligand-dependent nuclear hormone receptor transcription factor. The binding of a ligand (17β-estradiol) to ERα recruits estrogen response element to ERα [41]. ERα ligands, such as 17β-estradiol, induce the phosphorylation of Serine 118 of ERα [42]. To identify the potential of compound **1** as ERα and ERβ ligands, we carried out western blotting to evaluate ERα and ERβ activation by compound **1**. 17β-Estradiol (1 μM), which was a positive control, increased the phosphorylation of ERα to 5.88 ± 0.3-fold (*p* < 0.001) compared with untreated cells. Similarly, compound **1** increased ERα phosphorylation (50 μM: 2.55 ± 0.4, *p* < 0.05, 100 μM: 5.68 ± 0.7, *p* < 0.001; Figure 6). However, compound **1** and 17β-Estradiol (1 μM) not increased the phosphorylation of ERβ compared with untreated cells (Figure 6).

The results indicate that compound **1** induces phosphorylation of serine residues of ERα; thus, compound **1** may be considered an ERα ligand. Compound **1**, which contains phenolic hydroxyl groups similar to 17β-estradiol, seems to result in the activation of ERα via interactions with the hydroxyl groups and ligand-binding domains of ERα. 

### 2.5. Anti-Proliferative Effect of Extracts and Compounds from S. anglica on MCF-7 Cells

Breast cancer is divided into ER-positive and ER-negative, depending on the presence or absence of ER expression. More than 70% of breast cancers are ER-positive, and high estrogen levels have a close impact. High levels of estrogen in the blood increase the risk of breast cancer development. Moreover, HRT, which involves the administration of female hormones to compensate for the lack of estrogen in menopausal women, increases the risk of breast cancer development [43,44]. HRT acts as a ligand in ER-positive breast cancer cells and increases cell proliferation. 

Several previous studies on the safety of estrogen-like active compounds regarding ER-positive MCF-7 breast cancer cell proliferation under normal conditions have been reported [45,46,47]. Therefore, to evaluate the safety regarding the adverse effects associated with HRT treatment, the cytotoxic effects of extracts and compounds from *S. anglica* were measured in MCF-7 breast cancer cells using a cell viability assay. 

Therefore, we investigated the proliferation of ER-positive MCF-7 human breast cancer cells under normal conditions. None of the extracts and compounds from the belowground and aerial parts of *S. anglica* resulted in an increase in cell proliferation. Furthermore, *p*-hydroxybenzaldehyde (**3**) and *N*-*trans*-feruloyltyramine (**4**) reduced MCF-7 cell proliferation (Figure 7). The results indicate that extracts of *S. anglica* and its compounds have estrogenic effects, without increasing the risk of breast cancer development.

## 3. Materials and Methods

### 3.1. General Experimental Procedures

Optical rotation was measured using a Jasco DIP-1000 polarimeter (JASCO Inc., Tokyo, Japan) coupled with a sodium lamp (589 nm). Vacuum liquid chromatography (VLC) was performed using Silica Gel 60 (230–400 mesh, Merck KGaA, Darmstadt, Germany). LR-ESI-MS was performed on an Agilent 1260 series HPLC system coupled with a 6120 series single quadrupole mass spectrometer (Agilent, Santa Clara, CA, USA). HPLC was carried out on a Gilson HPLC system (321 pump, and UV/Vis-155, Gilson Inc., Middleton, WI, USA) and a Waters HPLC system (PDA 996, and 600 controllers, Waters Corporation, Milford, MA, USA). The 1D and 2D NMR spectra were obtained using a Bruker Avance DPX250 spectrometer (Bruker, Billerica, USA) and 600 MHz Fourier transform nuclear magnetic resonance spectrometer (VNS600, Agilent) at the Core Research Support Center for Natural Products and Medical Materials.

### 3.2. Plant Material 

*S. anglica* was collected from a mudflat on Ganghwa Island in South Korea in July 2016. The plant was identified by Dr. Hyukjae Choi, College of Pharmacy, Yeungnam University. The collected *S. anglica* was washed with tap water and dried at 24 °C in the shade. The dried plant specimen was deposited at the College of Pharmacy in Yeungnam University in South Korea. 

### 3.3. Extraction and Isolation of Compounds from S. anglica

The dried plant specimen was separated into two parts (aerial and belowground). A portion of the aerial (SA-a, 10 g) and belowground (SA-bg, 10 g) parts was extracted three times with four extraction solvents (100% deionized water (H_2_O), 30% ethanol in water, 50% ethanol in water, and 80% ethanol in water) at 20 ± 5 °C for 12 h to obtain the crude extracts of SA-bg-W (0.12 g), SA-bg-E30 (0.35 g), SA-bg-E50 (0.58 g), SA-bg-E80 (0.55 g), SA-a-W (0.92 g), SA-a-E30 (2.57 g), SA-a-E50 (1.34 g), and SA-a-E80 (3.30 g). 

The belowground (dry wt. 10 kg) parts of *S. anglica* was extracted with 50% ethanol in deionized water to obtain crude extracts of belowground parts (590 g) for compound isolation. 

A portion (150 g) of the 50% ethanolic extracts of the belowground parts was suspended in 1.0 L of deionized water and partitioned with hexanes, methylene chloride (MC), and ethyl acetate (EA) three times. Each layer was dried under reduced pressure to yield Hex-bg (472.5 mg), MC-bg (461.3 mg), and EA-bg (315.0 mg) fractions and a residual aqueous fraction (H_2_O-bg, 148 g). 

The EA-bg fraction was subjected to preparative HPLC using a reversed-phase C18 column (Phenomenex Luna C18, 100 Å, 22.5 × 250 mm) under the acidified solvent conditions (solvent A, 95% deionized water, 5% CH_3_CN, and 0.05% formic acid; solvent B 100% CH_3_CN and 0.05% formic acid; A:B = 80:20 for isocratic elution; flow rate = 6.0 mL/min; detection at 210 and 254 nm) to give a pure compound **1** (42.2 mg, t_R_ = 38.8 min). The other broad peak collection at the retention time range between 30.5 and 32.5 min was further purified by HPLC using a reversed-phase C18 column (Phenomenex Luna C18, 4.6 × 250 mm) under the following solvent conditions (solvent A, 95% deionized water, 5% CH_3_CN, and 0.05% formic acid; solvent B, 100% CH_3_CN and 0.05% formic acid; A:B = 85:15 for isocratic elution; flow rate = 1.0 mL/min) to obtain compound **2** (1.7 mg, t_R_ = 26.4 min). 

The MC-bg (461.3 mg) fraction was subjected to NP VLC and eluted with MC/MeOH to obtain seven fractions (MC-bg-A–MC-bg-G). The separation of MC-bg-D (96.5 mg) using preparative HPLC (Phenomenex Luna C18, 100 Å, 22.5 × 250 mm) under the following conditions: solvent A (100% deionized water), solvent B (100% CH_3_CN); A:B = 74:26; flow rate = 6.0 mL/min; detection at 210 and 254 nm afforded compound **3** (0.6 mg, t_R_ = 13.8 min) and compound **4** (5.5 mg, t_R_ = 35.6 min). MC-bg-F (85.2 mg) was separated by HPLC using a semi-preparative column (10.0 × 250 mm, 5 μm) under a B15% isocratic condition (solvent A, 95% deionized water, 5% CH_3_CN, and 0.05% formic acid; solvent B, 100% CH_3_CN and 0.05% formic acid) at a flow rate of 2.0 mL/min for 60 min to yield compound **5** (1.4 mg, t_R_ = 32.5 min).

### 3.4. MCF-7 Culture

The MCF-7 human breast cancer cell line was obtained from the American Type Culture Collection (Bethesda, MD, USA). Cells were maintained at 37 °C in a humidified, 5% CO_2_ atmosphere in Roswell Park Memorial Institute 1640 medium (RPMI 1640; Corning, Manassas, VA, USA) supplemented with 10% fetal bovine serum (Atlas, Fort Collins, CO, USA) and penicillin/streptomycin (Gibco, Grand Island, NY, USA). 

### 3.5. Determination of Estrogenic Effect in MCF-7 Cells

Experiments were performed using an E-screen assay as previously described [31]. Briefly, MCF-7 cells were plated at a density of 1 × 10^4^ cells/well in 48-well plates, allowed to adhere overnight, and challenged with the indicated concentrations of samples in phenol red-free RPMI (Gibco, Carlsbad, CA, USA) supplemented with 10% charcoal–dextran stripped serum (Innovative Research, Peary CourtNovi, MI, USA) for 144 h. Queens One Tablet was used as positive control in this study, it is constituted 200 mg of Red clover extract in tablet (360 mg), and its active compounds are isoflavones. It is commonly used for improvement of the symptoms about menopause such as hot flush, night sweats, emotional lability, agitation and insomnia. To confirm the estrogenic effect, the cells were maintained in the presence or absence of the estrogen receptor antagonist ICI 182,780 (Tocris Bioscience, Bristol, UK). Thereafter, Ez-Cytox (Daeil Lab Service Co., Seoul, Korea) was added into each well and the cells were incubated for 1 h. Subsequently, the absorbance was measured using a microplate reader (SPARK 10M; Tecan, Männedorf, Switzerland) at a wavelength of 450 nm. Data are presented as the mean ± SEM of three independent experiments performed in triplicate. Cell viability is expressed as a percentage of that in the untreated group (100%).

### 3.6. Determination of Anti-Proliferative Effect in MCF-7 Cells

Experiments were conducted as previously described [48]. Briefly, MCF-7 cells were plated at a density of 1 × 10^4^ cells/well in 96-well plates, allowed to adhere overnight, and challenged with the indicated concentrations of samples for 24 h. Thereafter, Ez-Cytox was added into each well and the cells were incubated for 1 h. Subsequently, the absorbance was measured using a microplate reader (PowerWave XS; Bio-Tek Instruments, Winooski, VT, USA) at a wavelength of 450 nm. Data are presented as the mean ± SEM of three independent experiments performed in triplicate. Cell viability is expressed as a percentage of that in the untreated group (100%).

### 3.7. Detection of Protein Expression in MCF-7 Cells

Experiments were conducted as previously described [48]. Briefly, MCFs were plated at a density of 2 × 10^5^ cells/well in 6-well plates, allowed to adhere overnight, and challenged with the indicated concentrations of samples for 24 h. The levels of phospho-ERα (p-ERα), ERα, phosphor-ERβ (p-ERβ), ERβ and glyceraldehyde 3-phosphate dehydrogenase (GAPDH) in the cells were determined by western blotting. Cell lysates were extracted using radioimmunoprecipitation assay buffer (Tech & Innovation, Gangwon, Korea). Protein concentration was determined using the Pierce™ BCA Protein Assay Kit (Pierce, Rockford, IL, USA). Protein separation was conducted using sodium dodecyl sulfate-polyacrylamide gel electrophoresis. Thereafter, proteins were transferred to a polyvinylidene difluoride membrane (Merck Millipore, Darmstadt, Germany) and blocked with 5% skim milk. Afterward, the membrane was incubated with diluted primary antibodies against p-ERα, ERα, p-ERβ, ERβ, and GAPDH (Cell Signaling Technology, Danvers, MA, USA) overnight at 4 °C. After washing the membrane, it was incubated at room temperature with a secondary rabbit IgG antibody for 1 h. The protein signal was measured using SuperSignal^®^ West Femto Maximum Sensitivity Chemiluminescent Substrate (Pierce) and the Fusion Solo Chemiluminescence System (PEQLAB Biotechnologie GmbH, Erlangen, Germany). Protein expression was normalized to that of GAPDH (reference protein). The analysis was performed using a Fusion Solo Chemiluminescence System. Relative protein expression was calculated and compared with that in an untreated group using ImageJ software (Version 1.51J; National Institutes of Health, Bethesda, MD, USA). 

### 3.8. Animals

Crj:CD (SD) rats (female, 14-day-old pups) were obtained from DBL (Chungbuk, Korea). After adaptation for 1 week, the 21-day-old immature rats were individually housed in cages. The immature rats were weighed and randomly assigned to each group. Body weight was recorded daily throughout the study. The rats were provided free access to water and a commercial diet. The room was maintained at a temperature of 22 ± 2 °C and relative humidity of 50 ± 10% and artificially illuminated with a fluorescent lamp in a 12-h light/dark cycle. All animals were managed based on the animal experimental guidelines suggested by the Institutional Animal Care and Use Committee (GIACUC-R2019035, approval on 15 November 2019).

### 3.9. Uterotrophic Assay

The rat uterotrophic assay proposed by the OECD is a screening method for the determination of the estrogenic properties of endocrine-disrupting chemicals [49]. In the present study, to test the estrogenic properties of the sample, we performed an uterotrophic assay with immature rats. The rats were divided into three groups of three rats. Corn oil and water were administered to the normal group and estradiol in corn oil (0.01 mg estradiol/kg body weight in 4 mL corn oil/kg body weight) was injected subcutaneously in the back of the rats daily in the control group. In the SA-bg-E50 group, corn oil (4 mL/kg body weight) was injected subcutaneously and SA-bg-E50 was orally administered at a dosage of 200 mg/kg body weight per day for 3 days. The normal and control groups were orally administered water for 3 days. Rats were euthanized approximately 24 h after the final administration. During necropsy, the uterus was carefully removed and weighed without any attached fat or mesentery.

### 3.10. Statistical Analyses

All data are expressed as the mean ± standard error of mean (SEM). Statistical analyses were conducted using one-way analysis of variance (ANOVA) and Tukey’s post-test to evaluate differences among the various experimental groups (GraphPad Prism 7.0, GraphPad Software Inc., San Diego, CA, USA). *p* < 0.05, *p* < 0.01, and *p* < 0.001 were considered significant.

## 4. Conclusions

Menopause presents with symptoms including facial flushing, vaginal atrophy, and osteoporosis, which are caused by decreased estrogen production. HRT, which is the administration of female hormones, is utilized to treat estrogen reduction, but it has been reported to increase the risk of breast cancer development. Therefore, we aimed to evaluate the estrogenic activities of *S. anglica* and its constituents and identify potential candidates for the treatment of estrogen reduction without the risk of breast cancer. We evaluated the estrogenic effects of extracts of *S. anglica* and its compounds in MCF-7 breast cancer cells. SA-bg-E50 and 1,3-di-*O*-*trans*-feruloyl-(−)-quinic acid (**1**) had significant estrogenic activity. Furthermore, the administration of SA-bg-E50 increased the uterine weight compared with that in the normal group in an immature female rat model. Compound **1** was regarded as an ERα ligand that induced the phosphorylation of serine residues of ERα. Additionally, compound **1** did not affect MCF-7 cell proliferation. Consequently, compound **1**, as an ERα ligand, could be an active compound of *S. anglica* and have estrogenic effect without side effects, such as the developing risk of breast cancer.

## Figures and Tables

**Figure 1 marinedrugs-19-00210-f001:**
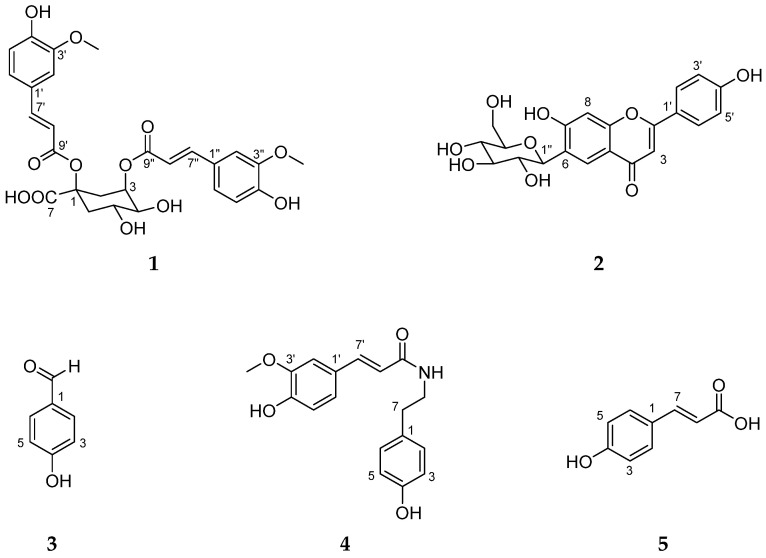
The structures of isolated compounds **1**–**5**.

**Figure 2 marinedrugs-19-00210-f002:**
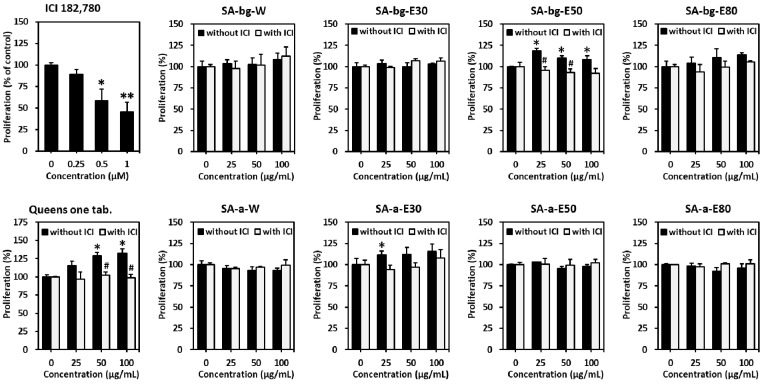
Estrogenic effects of the different extracts from belowground and aerial parts of *S. anglica* on MCF-7 cells. MCF-7 cells were challenged with 25, 50, and 100 μg/mL extracts in the presence or absence of 0.5 μM ICI 182,780 for 144 h in a charcoal–dextran stripped media condition. Proliferation was determined using an EZ-Cytox kit. Data are presented as mean ± standard error of mean (SEM) of at least three independent experiments. * *p* < 0.05 versus untreated cells. ** *p* < 0.01 versus untreated cells. # *p* < 0.05 versus each group without ICI 182,780. (SA-bg-W: water extract from belowground part of *S. anglica*, SA-bg-E30: 30% ethanol extract from belowground part, SA-bg-E50: 50% ethanol extract from belowground part, SA-bg-E80: 80% ethanol extract from belowground part, SA-a-W: water extract from aerial part, SA-a-E30: 30% ethanol extract from aerial part, SA-a-E50: 50% ethanol extract from aerial part, SA-a-E80: 80% ethanol extract from aerial part).

**Figure 3 marinedrugs-19-00210-f003:**
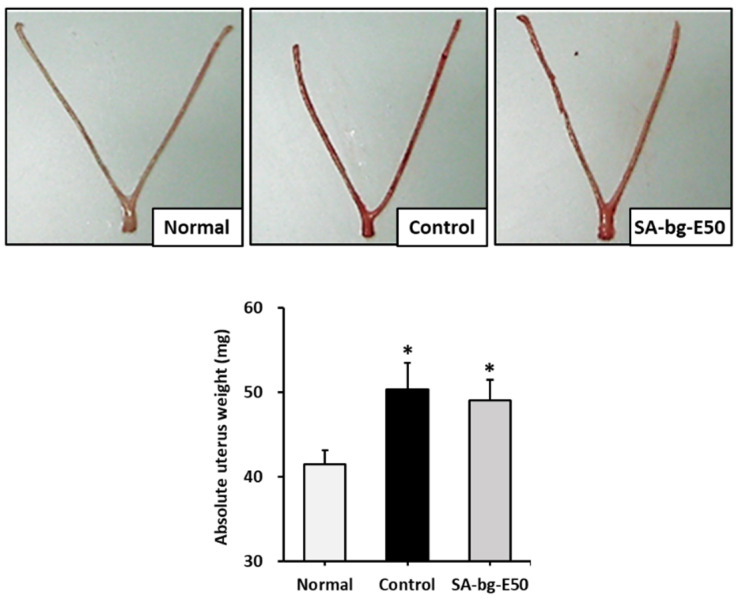
Uterotrophic activity of 50% ethanol extract from belowground part of *S. anglica* (SA-bg-E50) in the immature rat. 17β-Estradiol (10 μg/kg) was subcutaneously injected for 3 days. SA-bg-E50 (200 mg/kg) was orally administered for 3 days. Rats were euthanized 24 h after the final administration and the uterus was removed and weighed. Data are presented as mean ± SEM of at least three independent experiments. * *p* < 0.05 versus normal group. (Normal: corn oil and water treatment group, Positive control: 10 μg/kg 17β-estradiol injected group, SA-bg-E50: 50% ethanol extract from belowground part of *S. anglica* administered group).

**Figure 4 marinedrugs-19-00210-f004:**
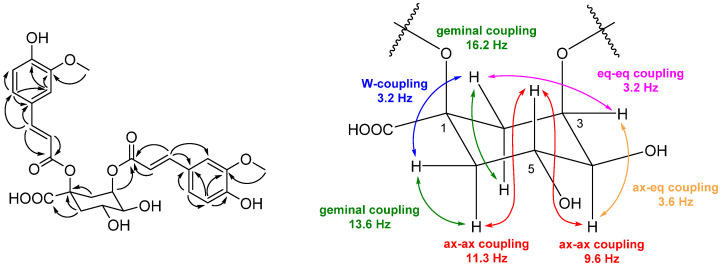
Key HMBC correlations and coupling constant analysis of the quinic acid unit of **1**.

**Figure 5 marinedrugs-19-00210-f005:**
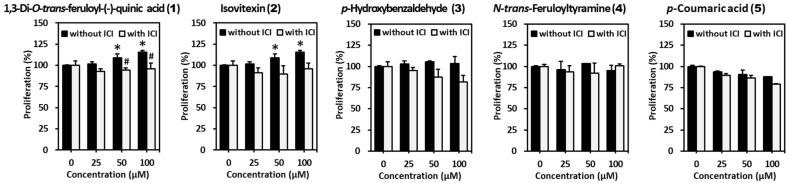
Estrogenic effects of compounds from belowground part of *S. anglica* on MCF-7 cells. MCF-7 cells were challenged with 25, 50, and 100 μM of compounds in the presence or absence of 500 nM ICI 182,780 for 144 h in a charcoal–dextran stripped media condition. Proliferation was determined using the EZ-Cytox kit. Data are presented as mean ± SEM of at least three independent experiments. * *p* < 0.05 versus untreated cells. # *p* < 0.05 versus each group without ICI 182,780.

**Figure 6 marinedrugs-19-00210-f006:**
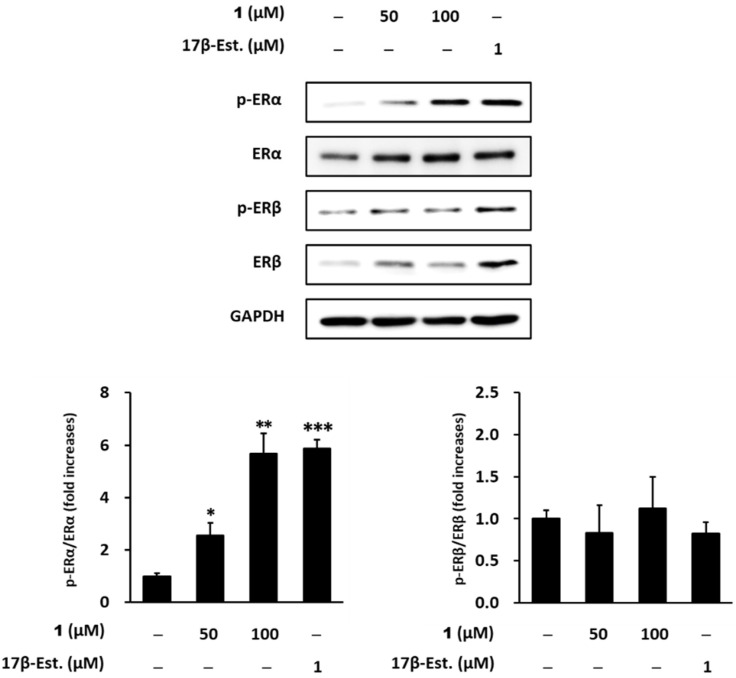
Protein levels of phospho-estrogen receptor-α (Ser 118), estrogen receptor-α, phospho-estrogen receptor-β (Ser 105), estrogen receptor-β and GAPDH against 1,3-di-*O*-*trans*-feruloyl-(−)-quinic acid (**1**)-treated MCF-7 cells. MCF-7 cells were challenged with 50 and 100 μM compound **1** for 24 h in a charcoal–dextran stripped media condition. 17β-Estradiol (17β-Est.) was used as a positive control. Subsequently, the protein levels were determined by western blotting. Data are presented as mean ± SEM of at least three independent experiments. * *p* < 0.05, ** *p* < 0.01, and *** *p* < 0.001 versus untreated cells.

**Figure 7 marinedrugs-19-00210-f007:**
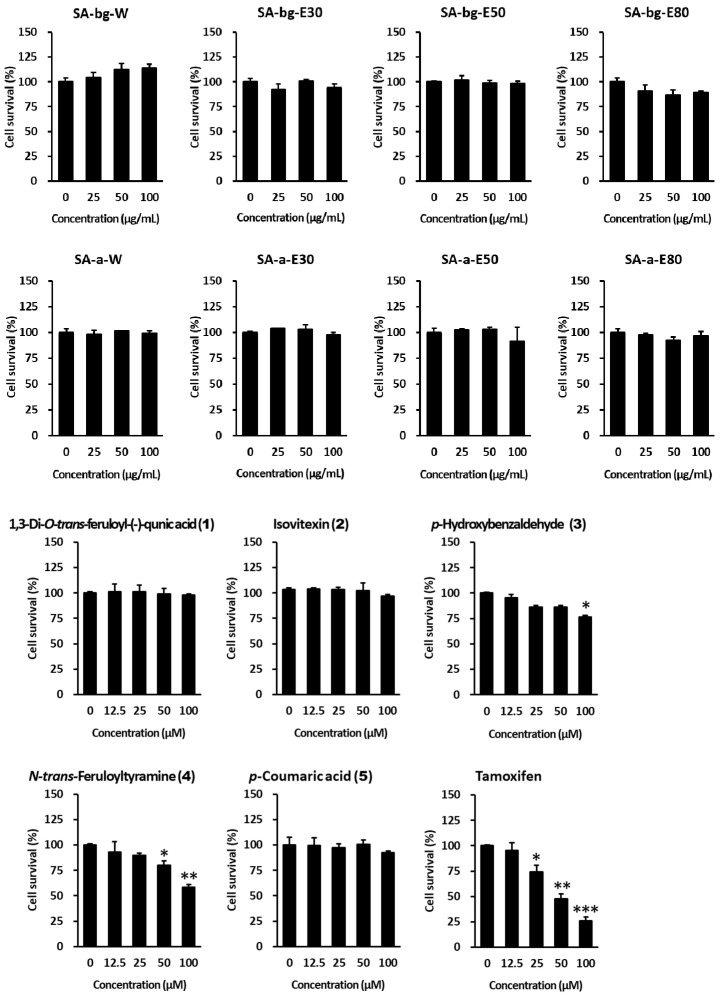
Anti-proliferative effects of the different extracts and compounds from belowground and aerial parts of *S. anglica* on MCF-7 cells. MCF-7 cells were challenged with 25, 50, and 100 μg/mL extracts and 12.5, 25, 50, and 100 μM of compounds for 24 h in a normal media condition. Tamoxifen used as positive control. Proliferation was determined using the EZ-Cytox kit. Data are presented as mean ± SEM of at least three independent experiments. ** p* < 0.05, *** p* < 0.01 and **** p* < 0.001 versus untreated cells. (SA-bg-W: water extract from belowground parts of *S. anglica*, SA-bg-E30: 30% ethanol extract from belowground parts of *S. anglica*, SA-bg-E50: 50% ethanol extract from belowground parts of *S. anglica*, SA-bg-E80: 80% ethanol extract from belowground parts of *S. anglica*, SA-a-W: water extract from aerial parts of *S. anglica*, SA-a-E30: 30% ethanol extract from aerial parts of *S. anglica*, SA-a-E50: 50% ethanol extract from aerial parts of *S. anglica*, SA-a-E80: 80% ethanol extract from aerial parts of *S. anglica*).

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
