# Peer review of "Estrogenic Effects of Extracts and Isolated Compounds from Belowground and Aerial Parts of Spartina anglica"

_marinedrugs, 2021, doi:10.3390/md19040210_

Round 1

Reviewer 1 Report

In this report, the authors investigate estrogenic effects of the extracts of Spartina anglica. S. anglica disturb the mudflat ecosystem and this plant is designated one of the “unintroduced species” in Korea. I think the concept is very interesting and important to sustainable development in the medical and environmental aspects. However, this study should be revised clearly in the effectiveness of S. anglica extracts to relieve menopausal symptoms.

â—‹Major Comments

  1. 2.1. Estrogenic Effects of Different Extracts from Belowground and Aerial Parts of S. anglica on MCF-7 Cells (Figure 2)

Page 4, line 109–112: Effects of SA-bg-E50 on proliferation of MCF-7 cells were observed. However, these effects do not correlate with concentration of SA-bg-E50. The authors should discuss the reason why the estrogenic effects were not concentration dependent.

  1. 2.4. Estrogenic Effects of Compounds Isolated from Belowground and Aerial Parts of S. anglica on MCF-7 cells

Why did the authors isolate these compounds from 80% ethanol soluble fraction of aerial parts of S. anglica? If SA-bg-50 have estrogenic effects, I want to extract compounds from 50% ethanol soluble fraction of belowground parts of S. anglica.

  1. 2.4. Estrogenic Effects of Compounds Isolated from Belowground and Aerial Parts of S. anglica on MCF-7 cells (Figure 5)

Page 6, line 204–206: I think the Compound (1) and (2) showed almost same effects on MCF-7 cells. Was the sample size appropriate?

  1. 2.5 Anti-proliferative Effect of Extracts and Compounds from S. anglica on MCF-7 Cells

In Figures 6 and 7, the authors confirmed the effects of Compound (1) on same cell line. Although Compound (1) induced phosphorylation of ERα in Figure 6, estrogenic effects of Compound (1) were not observed in Figure 7. To detect the effects of normal media condition, the authors should add the results of positive control (such as 17β-Estradiol) in Figure 7.

â—‹Minor Comments

  1. Page 5, line 141: I think “The results indicate that SA-a-E50” should be “The results indicate that SA-bg-E50.”
  2. Page 6, line 199–200: The authors should confirm the concentration of Compounds (μg/mL or μM).
  3. Page 8, Figure 7: Why are there the two different 25 μM bar graphs ?
  4. Page 8, line 250: The authors should delete ***p< 0.001. This symbol was not used in Figure 7.
  5. Page 8, line 259: I think Figure 6 was not relevant this sentence.
  6. Page 9, line 313: The authors should unify abbreviation of ethyl acetate (EA or EtOAc).
  7. Page 10, line 342 and 350: If the data is presented as mean ± SEM, the authors should change these sentences.
  8. Page 10, line 362–365: The authors should confirme temperature for primary antibody reaction (room temperature or 4°C).
  9. Page 11, line 398–400: The authors should add software name (maybe Prism).

Author Response

Response to Comments of Reviewer #1

All the authors deeply appreciated the reviewer’s valuable comments to improve the quality of our manuscript. We revised the manuscript as the reviewer suggested. Also, all the mistakes and typos found during the inspection of the manuscript were all revised.

In this report, the authors investigate estrogenic effects of the extracts of Spartina anglicaS. anglica disturb the mudflat ecosystem and this plant is designated one of the “unintroduced species” in Korea. I think the concept is very interesting and important to sustainable development in the medical and environmental aspects. However, this study should be revised clearly in the effectiveness of S. anglica extracts to relieve menopausal symptoms.

Major Comments

  1. 1. Estrogenic Effects of Different Extracts from Belowground and Aerial Parts of S. anglicaon MCF-7 Cells (Figure 2)

Page 4, line 109–112: Effects of SA-bg-E50 on proliferation of MCF-7 cells were observed. However, these effects do not correlate with concentration of SA-bg-E50. The authors should discuss the reason why the estrogenic effects were not concentration dependent.

: Thank you for helpful comments. As suggested, we added a short description on the reason why the estrogenic effects of SA-bg-E50 were not concentration dependent in lines 112-114. The extract is a complex of various phytochemicals. If there are both estrogenic phytochemicals and their competitive inhibitors in the test sample, the estrogenic activity of test samples can be concentrate-independent. The manuscript is revised as follows:

“It seems that SA-bg-E50 contains the estrogenic phytochemicals along with their competitive inhibitors. These two opposite actions can cause concentration independent estrogenic activities of SA-bg-E50.”

  1. 2.4. Estrogenic Effects of Compounds Isolated from Belowground and Aerial Parts of S. anglicaon MCF-7 cells

Why did the authors isolate these compounds from 80% ethanol soluble fraction of aerial parts of S. anglica? If SA-bg-50 have estrogenic effects, I want to extract compounds from 50% ethanol soluble fraction of belowground parts of S. anglica.

: We apologize for this inadvertent error. In the carful inspection of the manuscript for the clarification on the issues raised by reviewer, we found mistakes on the description of phytochemical isolation from S. anglica extracts. In this study, we isolated compounds from the 50% ethanol soluble fraction of belowground part of S. anglica which was selected from the most estrogenic among the tested extracts. Therefore, we revised the part of “3.3 Extraction and Isolation of compounds from S. anglica”. 

  1. 4. Estrogenic Effects of Compounds Isolated from Belowground and Aerial Parts ofS. anglica on MCF-7 cells (Figure 5)

Page 6, line 204–206: I think the Compound (1) and (2) showed almost same effects on MCF-7 cells. Was the sample size appropriate?

: Thank you for the sharp comments on the data description. The description on the estrogenic activity of compound 2 was inconsistent with the figure 5. We inspected the estrogenic activity data and correct the description on the activity in Lines 204-207.

Isovitexin (2) also increased cell proliferation in a concentration-dependent manner (50 μM: 106.9 ± 4.1%, 100 μM: 111.2 ± 1.9%). Cell proliferation was suppressed to 89.8 ± 9.9% (50 μM) and 93.4 ± 7.1% (100 μM) by ICI 182,780, however it was not significant. (Figure 5).”

In addition, the unit of tested samples were corrected. As shown in Figure 5, the estrogenic activity of compounds 1-5 were measured at the sample concentrations of 25, 50, and 100 μM.

The results were obtained from the three independent experiments and we think the sample size was appropriate.

  1. 5 Anti-proliferative Effect of Extracts and Compounds from S. anglica on MCF-7 Cells

In Figures 6 and 7, the authors confirmed the effects of Compound (1) on same cell line. Although Compound (1) induced phosphorylation of ERα in Figure 6, estrogenic effects of Compound (1) were not observed in Figure 7. To detect the effects of normal media condition, the authors should add the results of positive control (such as 17β-Estradiol) in Figure 7.

: We appreciate for helpful comments. According to your comments, we added results of positive control in Figure 7.

 - please check the attached file. 

Figure 7. Anti-proliferative effects of the different extracts and compounds from belowground part of S. anglica on MCF-7 cells. MCF-7 cells were challenged with 25, 50, and 100 μg/mL extracts and 25, 50, and 100 μM of compounds for 24 h in a normal media condition. Tamoxifen used as positive control. Proliferation was determined using the EZ-Cytox kit. Data are presented as mean ± SEM of at least three independent experiments. *p < 0.05, **p < 0.01 and ***p < 0.001 versus untreated cells. (SA-bg-W: water extract from belowground parts of S. anglica, SA-bg-E30: 30% ethanol extract from belowground parts of S. anglica, SA-bg-E50: 50% ethanol extract from belowground parts of S. anglica, SA-bg-E80: 80% ethanol extract from belowground parts of S. anglica, SA-a-W: water extract from aerial parts of S. anglica, SA-a-E30: 30% ethanol extract from aerial parts of S. anglica, SA-a-E50: 50% ethanol extract from aerial parts of S. anglica, SA-a-E80: 80% ethanol extract from aerial parts of S. anglica).

 Minor Comments

  1. Page 5, line 141: I think “The results indicate that SA-a-E50” should be “The results indicate that SA-bg-E50.” 

: Thank you for the correction on the manuscript. As suggested, the phrase was revised in the newly submitted manuscript. Now it reads as follows: “The results indicate that SA-bg-E50”

  1. Page 6, line 199–200: The authors should confirm the concentration of Compounds (μg/mL or μM). 

: We thank for your valuable comments and apologize for incorrect concentration unit. According to your comments, we revised concentration unit as shown in Lines 201-202. “(50 μM: 109.0 ± 4.7%, 100 μM: 115.4 ± 2.1%)”

  1. Page 8, Figure 7: Why are there the two different 25 μM bar graphs?

: We apologize for incorrect concentration in graphs. We revised concentration unit as shown in Figure 7.

  1. Page 8, line 250: The authors should delete ***p< 0.001. This symbol was not used in Figure 7.

: Newly added data on the cell proliferation by tamoxifen showed “***” and the symbol was still included in the legend of Figure 7.

  1. Page 8, line 259: I think Figure 6 was not relevant this sentence.

: Page 8, line 270 “Figures 6 and 7” was revised as “Figure 7”.

  1. Page 9, line 313: The authors should unify abbreviation of ethyl acetate (EA or EtOAc).

: We unified abbreviation of ethyl acetate as EA and added description of abbreviation in page 9, line 302.

  1. Page 10, line 342 and 350: If the data is presented as mean ± SEM, the authors should change these sentences.

: “S.D.” was revised as “SEM” at lines 345 and 354 of page 10.

  1. Page 10, line 362–365: The authors should confirm temperature for primary antibody reaction (room temperature or 4°C).

: The reaction was performed at the temperature of 4°C. The sentence was revised as shown in Line 366-368.

“Afterward, the membrane was incubated with diluted primary antibodies against p-ERα, ERα, p-ERβ, ERβ,and GAPDH (Cell Signaling Technology, Danvers, MA, USA) overnight at 4 °C.”

  1. Page 11, line 398–400: The authors should add software name (maybe Prism).

: We added software name “Prism 7.0” in line 402.

List of Changes in the Revised Manuscript

- In page 1, lines 8 and 16: The e-mail addresses of two authors, Geum Jin Kim and Sang Hee Shim, were changed as their institutional email addresses in the newly submitted manuscript.

- In page 1, line 24: The phrase “menopausal symptoms” was revised as “estrogen reduction”.

- In page 1, line 26: The phrase “a uterotrophic” was corrected as “an uterotrophic”.

- In page 1, line 27: The description “and aerial parts” was changed to “part”.

- In page 1, lines 32: The phrase “enhance various menopausal symptoms” was changed as “have estrogenic effects”.

- In page 2, line 83: The phrase “menopausal symptoms” was revised as “estrogen reduction”.

- In page 3, line 100: Effect of ICI 182,780 on MCF-7 cell proliferation was added to Figure 2.

- In page 3, line 102: The unit of ICI 182,780 was changed to micromolar.

- In page 4, lines 112-114: A short description on the concentration independent estrogenic activity by SA-bg-E50 was added.

- In page 4, line 128: A description “relieve menopausal symptoms” was modified as “have estrogenic effects”

- In page 5, lines 142-143: Unclear description was revised and now it reads as follows: The results indicate that SA-bg-E50 hypertrophies the uterus owing to its estrogenic activity (Figure 3).

- In page 5, line 145: The style of compound number was revised as bold style. 

- In page 6, line 197: The description of “Belowground and Aerial Parts” was changed to “Belowground Part”.

- In page 6, line 199-200: The phrase “belowground and aerial parts” was revised as “belowground part”.

- In page 6, lines 200, 203, 204, 207 and 209: The format of compound number was revised as bold style.

- In page 6, lines 201 and 205: The units of test samples were corrected as “μM”.

- In page 6, lines 204-208: The descriptions on estrogenic activities of compounds 1 and 2 were revised.

- In page 6, line 211: In the description of figure 6, “belowground and aerial parts” was corrected as “belowground part”.

- In page 6, lines 219-220: A sentence describing the additional experiments on ERβ was add.

- In page 7, lines 224-225: The descriptive sentence on the phosphorylation on ERβ by compound 1 and 17β-estradiol treatment was added.

- In page 7, lines 227-234: Bar chart for the phosphorylation of ERβ by compound 1 and 17β-estradiol treatment was added. The description of figure 6 was revised as the figure changed.

- In page 7, line 238: The sentence “Thus, compound 1 is expected to relieve various menopausal symptoms.” was removed.

- In page 7, line 245: Two references supporting side effects of HRT were added.

- In page 7, lines 247-251: A paragraph describing the breast cancer development by estrogen-like compounds along with references were added.

- In page 8, line 253: A cell proliferation data of tamoxifen was added and the test concentrations of compounds 1-5 were corrected.

- In page 8, lines 254-264: In the description of figure 7, the descriptions were revised as the figure modified.

- In page 8, line 270: The phrase “relieve various menopausal symptoms” was revised as “have estrogenic effects”.

- In pages 9-10, lines 296-323: Descriptions on the isolation of compounds 1-5 were revised.

- In page 10, lines 335-339: Description on Queens one tablet was added as requested.

- In page 10, lines 344 and 353: “S.D.” was corrected as “SEM”.

- In page 10, lines 348, 356: The reference number changed from 43 to 48.

- In page 10, line 358-359: The additional experiment related with ERβ was described additionally.

- In page 10, lines 366-368: Experimental condition was corrected to prevent unnecessary confusion.

- In page 11, line 388: The reference number 44 was revised to 48.

- In page 11, line 389: The phrase “a uterotrophic” was changed to “an uterotrophic”.

- In page 11, line 402: The detail on graphic software was added.

- In page 11, lines 407 and 410: The term “menopausal symptoms” was corrected to be “estrogen reduction”.

- In page 11, lines 416-418: The final conclusion on the potential of compound 1 was revised.

- In pages 13-14, lines 506-514: Five references were added in the reference list.

- In page 14, lines 515-518: The reference number 43 and 44 was changed to 48 and 49.

List of Changes in the Supplementary Materials

- The e-mail addresses of two authors, Geum Jin Kim and Sang Hee Shim, were changed as their institutional email addresses.

Reviewer 2 Report

The authors studied the estrogenic effects of extracts and isolated compounds from belowground and aerial parts of Spartina anglica in MCF-7 breast cancer cells and in immature female rats. The main conclusion of this article is that two compounds found in Spartina anglica had significant estrogenic activity. SA-bg-E50 increased the uterine weight in the rat model, while 1,3-di-O-trans-feruloyl-(-)-quinic acid (Compound 1) was regarded as an ERα ligand that induced the phosphorylation of serine residues in this estrogen receptor, without increasing cell proliferation. The authors indicate this compound 1 as a possible treatment against menopausal symptoms.

The study is well designed, and the overall paper is interesting; however, based on the complexity of the results, there are some points that should be considered:

  1. Despite ERα is the main estrogen receptor found un MCF-7 cells, another estrogen receptor, ERβ, is also found in this cell line. The fact that Compound 1 produced no significant differences in cell proliferation could be due, at least in part, to the presence of ERβ since this receptor has antiproliferative and cytostatic effects. In my opinion, authors should complete this study analyzing the levels of ERβ and modulating its expression with, for example, a specific siRNA.
  2. In addition to my comment about ERβ, I would like to see the proliferation of cells treated with ICI in comparison with the cells without ICI. It is important to know how ICI is affecting to cell viability in basal conditions to be able to compare between experimental groups. * In the article both conditions in 0 µg/µl or 0 µM treatment are set at 100% in the proliferation figures.
  3. Finally, the conclusion regarding the Compound 1 is too bold considering the results obtained. More studies focusing on the physiology of menopausal symptoms would be necessary to support this conclusion.

Author Response

Response to Comments of Reviewer #2

All the authors deeply appreciated the reviewer’s valuable comments and suggestions which is helpful to improve the quality of our manuscript. We revised the manuscript as the reviewer suggested. Also, all the mistakes and typos found during the inspection of the manuscript were all revised.

The authors studied the estrogenic effects of extracts and isolated compounds from belowground and aerial parts of Spartina anglica in MCF-7 breast cancer cells and in immature female rats. The main conclusion of this article is that two compounds found in Spartina anglica had significant estrogenic activity. SA-bg-E50 increased the uterine weight in the rat model, while 1,3-di-O-trans-feruloyl-(-)-quinic acid (Compound 1) was regarded as an ERα ligand that induced the phosphorylation of serine residues in this estrogen receptor, without increasing cell proliferation. The authors indicate this compound 1 as a possible treatment against menopausal symptoms.

The study is well designed, and the overall paper is interesting; however, based on the complexity of the results, there are some points that should be considered:

  1. Despite ERα is the main estrogen receptor found un MCF-7 cells, another estrogen receptor, ERβ, is also found in this cell line. The fact that Compound 1 produced no significant differences in cell proliferation could be due, at least in part, to the presence of ERβ since this receptor has antiproliferative and cytostatic effects. In my opinion, authors should complete this study analyzing the levels of ERβ and modulating its expression with, for example, a specific siRNA.

: We appreciate for helpful comments. According to your comments, we added the levels of ERβ in Figure 6.

 - Please check the attachment

Figure 6. Protein levels of phospho-estrogen receptor-α (Ser 118), estrogen receptor-α, phospho-estrogen receptor-β (Ser 105), estrogen receptor-β and GAPDH against 1,3-di-O-trans-feruloyl-(-)-quinic acid (1)-treated MCF-7 cells. MCF-7 cells were challenged with 50 and 100 μM compound 1 for 24 h in a charcoal–dextran stripped media condition. 17β-Estradiol (17β-Est.) was used as a positive control. Subsequently, the protein levels were determined by western blotting. Data are presented as mean ± SEM of at least three independent experiments. *p < 0.05, **p < 0.01, and ***p < 0.001 versus untreated cells.

  1. In addition to my comment about ERβ, I would like to see the proliferation of cells treated with ICI in comparison with the cells without ICI. It is important to know how ICI is affecting to cell viability in basal conditions to be able to compare between experimental groups. * In the article both conditions in 0 µg/µl or 0 µM treatment are set at 100% in the proliferation figures.

: The effect of cell proliferation by ICI 182,780 were measured in MCF-7 breast cancer cells using a cell viability assay. Cell proliferation was significantly suppressed to 58.8 ± 13.1% (500 nM) and 45.2 ± 11.4% (1000 nM) by ICI 182,780. The following image was included in Figure 2.

The effect of MCF-7 cell proliferation by ICI 182,780

  1. Finally, the conclusion regarding the Compound 1 is too bold considering the results obtained. More studies focusing on the physiology of menopausal symptoms would be necessary to support this conclusion.

: As suggested by you, we revised conclusion in line 417-419.

“Consequently, compound 1, as an ERα ligand, could be an active compound of S. anglica and have estrogenic effect without side effects, such as the developing risk of breast can-cer.“

List of Changes in the Revised Manuscript

- In page 1, lines 8 and 16: The e-mail addresses of two authors, Geum Jin Kim and Sang Hee Shim, were changed as their institutional email addresses in the newly submitted manuscript.

- In page 1, line 24: The phrase “menopausal symptoms” was revised as “estrogen reduction”.

- In page 1, line 26: The phrase “a uterotrophic” was corrected as “an uterotrophic”.

- In page 1, line 27: The description “and aerial parts” was changed to “part”.

- In page 1, lines 32: The phrase “enhance various menopausal symptoms” was changed as “have estrogenic effects”.

- In page 2, line 83: The phrase “menopausal symptoms” was revised as “estrogen reduction”.

- In page 3, line 100: Effect of ICI 182,780 on MCF-7 cell proliferation was added to Figure 2.

- In page 3, line 102: The unit of ICI 182,780 was changed to micromolar.

- In page 4, lines 112-114: A short description on the concentration independent estrogenic activity by SA-bg-E50 was added.

- In page 4, line 128: A description “relieve menopausal symptoms” was modified as “have estrogenic effects”

- In page 5, lines 142-143: Unclear description was revised and now it reads as follows: The results indicate that SA-bg-E50 hypertrophies the uterus owing to its estrogenic activity (Figure 3).

- In page 5, line 145: The style of compound number was revised as bold style. 

- In page 6, line 197: The description of “Belowground and Aerial Parts” was changed to “Belowground Part”.

- In page 6, line 199-200: The phrase “belowground and aerial parts” was revised as “belowground part”.

- In page 6, lines 200, 203, 204, 207 and 209: The format of compound number was revised as bold style.

- In page 6, lines 201 and 205: The units of test samples were corrected as “μM”.

- In page 6, lines 204-208: The descriptions on estrogenic activities of compounds 1 and 2 were revised.

- In page 6, line 211: In the description of figure 6, “belowground and aerial parts” was corrected as “belowground part”.

- In page 6, lines 219-220: A sentence describing the additional experiments on ERβ was add.

- In page 7, lines 224-225: The descriptive sentence on the phosphorylation on ERβ by compound 1 and 17β-estradiol treatment was added.

- In page 7, lines 227-234: Bar chart for the phosphorylation of ERβ by compound 1 and 17β-estradiol treatment was added. The description of figure 6 was revised as the figure changed.

- In page 7, line 238: The sentence “Thus, compound 1 is expected to relieve various menopausal symptoms.” was removed.

- In page 7, line 245: Two references supporting side effects of HRT were added.

- In page 7, lines 247-251: A paragraph describing the breast cancer development by estrogen-like compounds along with references were added.

- In page 8, line 253: A cell proliferation data of tamoxifen was added and the test concentrations of compounds 1-5 were corrected.

- In page 8, lines 254-264: In the description of figure 7, the descriptions were revised as the figure modified.

- In page 8, line 270: The phrase “relieve various menopausal symptoms” was revised as “have estrogenic effects”.

- In pages 9-10, lines 296-323: Descriptions on the isolation of compounds 1-5 were revised.

- In page 10, lines 335-339: Description on Queens one tablet was added as requested.

- In page 10, lines 344 and 353: “S.D.” was corrected as “SEM”.

- In page 10, lines 348, 356: The reference number changed from 43 to 48.

- In page 10, line 358-359: The additional experiment related with ERβ was described additionally.

- In page 10, lines 366-368: Experimental condition was corrected to prevent unnecessary confusion.

- In page 11, line 388: The reference number 44 was revised to 48.

- In page 11, line 389: The phrase “a uterotrophic” was changed to “an uterotrophic”.

- In page 11, line 402: The detail on graphic software was added.

- In page 11, lines 407 and 410: The term “menopausal symptoms” was corrected to be “estrogen reduction”.

- In page 11, lines 416-418: The final conclusion on the potential of compound 1 was revised.

- In pages 13-14, lines 506-514: Five references were added in the reference list.

- In page 14, lines 515-518: The reference number 43 and 44 was changed to 48 and 49.

List of Changes in the Supplementary Materials

- The e-mail addresses of two authors, Geum Jin Kim and Sang Hee Shim, were changed as their institutional email addresses.

Reviewer 3 Report

Natural or phytochemical hormones are of interests in female reproductive health and disorders. Environmental estrogens are known to cause human reproductive dysfunction due to their long and consistent exposure, while herbal estrogens can be potentially used for the treatments of estrogen deficiencies. This study evaluated the estrogenic activities of Spartina anglica and the isolated compounds by using estrogen receptor positive breast cancer cells and rat model, to elucidate the potential beneficial effect of Spartina anglica on relieving menopausal symptoms without causing breast cancer risk. The results are of interest, however, there are several non-negligible issues regarding experimental design and result interpretation.

The major concern of this study is the animal model for menopause study. The authors used immature female rat model to mimic the menopausal symptoms, although the immature rat has very low estrogen level, however, the physiological status is totally different with menopausal status. Is there any reference to support the use of immature rat for menopausal studies?

The authors indicated that the “changes in thickening are observed in the photograph”, which is not appropriate. Increased estrogen level or estrogen treatments can cause uterine edema and hyperaemia, the thickness of uterus should be determined by measuring the thickness of endometrium and myometrium with HE staining.

Page 5, line 141, the statement of “relieves menopausal symptoms” based on Figure 3 is not appropriate. Increased uterine thickness and weight are not representative for menopausal symptoms. In fact, the authors used "relieves menopausal symptoms" across the manuscript, however, they didn't investigate the menopausal symptoms. This statement should be corrected in this study.  

This reviewer is confused by the discrepant results between Figure 2, Figure 5, and Figure 7. In figure 2, SA-bg-E50 and SA-a-E30 showed pro-proliferation effect on MCF-7 cells, while in Figure 7, both of them had no effect on MCF-7 proliferation. In Figure 5, compound (1) and (2) significantly stimulates MCF-7 proliferation, while they had no effect in Figure 7.  

The authors need to provide the detailed information of Queens One tab, including active reagent, dosage, manufactural information.

For compounds 1-5, are they all existed in both of SA-bg and SA-a?

Author Response

Response to Comments of Reviewer #3

Natural or phytochemical hormones are of interests in female reproductive health and disorders. Environmental estrogens are known to cause human reproductive dysfunction due to their long and consistent exposure, while herbal estrogens can be potentially used for the treatments of estrogen deficiencies. This study evaluated the estrogenic activities of Spartina anglica and the isolated compounds by using estrogen receptor positive breast cancer cells and rat model, to elucidate the potential beneficial effect of Spartina anglica on relieving menopausal symptoms without causing breast cancer risk. The results are of interest, however, there are several non-negligible issues regarding experimental design and result interpretation.

The major concern of this study is the animal model for menopause study. The authors used immature female rat model to mimic the menopausal symptoms, although the immature rat has very low estrogen level, however, the physiological status is totally different with menopausal status. Is there any reference to support the use of immature rat for menopausal studies?

: The rodent uterotrophic assay is most extensively used for screening estrogenic compounds. There are various protocols depending on animal (immature, hypophysectomized, or ovariectomized animal), and different routes of administration. As you mentioned above, although the immature rat has very low estrogen level, however, the physiological status is totally different with menopausal status. However, Kang et al. had been provided that immature uterotrophic assay is more sensitive than ovariectomized uterotrophic assay for the detection of estrogenic effect in Sprague–Dawley rats (1). And other researches also had been reported that immature uterotrophic assay used for the detection of estrogenic effect (2,3).

[Related reference]

  1. Kang, K.S., Kim, H.S., Ryu, D.Y., Che, J.H., & Lee, Y.S. (2000). Immature uterotrophic assay is more sensitive than ovariectomized uterotrophic assay for the detection of estrogenicity of p-nonylphenol in Sprague–Dawley rats. Toxicology letters118(1-2), 109-115.
  2. Heneweer, M., Houtman, R., Poortman, J., Groot, M., Maliepaard, C., & Peijnenburg, A. (2007). Estrogenic effects in the immature rat uterus after dietary exposure to ethinylestradiol and zearalenone using a systems biology approach. Toxicological sciences99(1), 303-314.
  3. Sun, L., Yu, T., Guo, J., Zhang, Z., Hu, Y., Xiao, X., ... & Li, J. (2016). The estrogenicity of methylparaben and ethylparaben at doses close to the acceptable daily intake in immature Sprague-Dawley rats. Scientific reports6(1), 1-9.

The authors indicated that the “changes in thickening are observed in the photograph”, which is not appropriate. Increased estrogen level or estrogen treatments can cause uterine edema and hyperaemia, the thickness of uterus should be determined by measuring the thickness of endometrium and myometrium with HE staining.

: We appreciate for valuable comments. Because the thickness of endometrium and myometrium were not measured in this study, we have removed inappropriate sentence to solve this discrepancy. The newly prepared paragraph (lines 140-143) is as follows:

“The uterine weight of the positive control group was significantly increased, to 50.3 ± 3.1 mg, compared with that of the normal group (41.5 ± 1.6 mg). After the administration of SA-bg-E50, the uterine weight was 49.0 ± 2.5 mg. The results indicate that SA-bg-E50 hypertrophies the uterus owing to its estrogenic activity (Figure 3).”

Page 5, line 141, the statement of “relieves menopausal symptoms” based on Figure 3 is not appropriate. Increased uterine thickness and weight are not representative for menopausal symptoms. In fact, the authors used "relieves menopausal symptoms" across the manuscript, however, they didn't investigate the menopausal symptoms. This statement should be corrected in this study.  

: We apologize for incorrect description. We have corrected the statement in Lines 142-143.

“The results indicate that SA-bg-E50 hypertrophies the uterus owing to its estrogenic activity (Figure 3).”

This reviewer is confused by the discrepant results between Figure 2, Figure 5, and Figure 7. In figure 2, SA-bg-E50 and SA-a-E30 showed pro-proliferation effect on MCF-7 cells, while in Figure 7, both of them had no effect on MCF-7 proliferation. In Figure 5, compound (1) and (2) significantly stimulates MCF-7 proliferation, while they had no effect in Figure 7.  

: E-screen assay was conducted to evaluate the estrogenic activity of the compounds for hormone replacement therapy (HRT). Although the same cells are used, anticancer effects of phytoestrogens in breast cancer is well known. The same experimental method that we used in the following papers has been applied and reviewed. We have revised the manuscript to solve this enormous discrepancy following reviewer’s indication (Line 248-252)

“Several previous studies on the safety of estrogen-like active compounds regarding ER-positive MCF-7 breast cancer cell proliferation under normal conditions have been reported. Therefore, to evaluate the safety regarding the adverse effects associated with HRT treatment, the cytotoxic effects of extracts and compounds from S. anglica were measured in MCF-7 breast cancer cells using a cell viability assay.”

[Related reference]

  1. Parrella, A., Lavorgna, M., Criscuolo, E., Russo, C., & Isidori, M. (2014). Estrogenic activity and cytotoxicity of six anticancer drugs detected in water systems. Science of the total environment, 485, 216-222.
  2. Hsieh, C. J., Hsu, Y. L., Huang, Y. F., & Tsai, E. M. (2018). Molecular mechanisms of anticancer effects of phytoestrogens in breast cancer. Current Protein and Peptide Science, 19(3), 323-332.
  3. Ahn, S. Y., Jo, M. S., Lee, D., Baek, S. E., Baek, J., Yu, J. S., ... & Kim, K. H. (2019). Dual effects of isoflavonoids from Pueraria lobata roots on estrogenic activity and anti-proliferation of MCF-7 human breast carcinoma cells. Bioorganic chemistry, 83, 135-144.

The authors need to provide the detailed information of Queens One tab, including active reagent, dosage, manufactural information.

: According your comments, information of Queens One tab was added in line 336-340.

“Queens One tab. was used as positive control in this study, it is constituted 200 mg of Red clover extract in tablet (360 mg), and its active compounds are isoflavones. It is commonly used for improvement of the symptoms about menopause such as hot flush, night sweats, emotional lability, agitation and insomnia.”

For compounds 1-5, are they all existed in both of SA-bg and SA-a?

: All the compounds 1-5 in this study were isolated from the SA-bg, but they are also found in the extract of the aerial parts of S. anglica extract.

List of Changes in the Revised Manuscript

- In page 1, lines 8 and 16: The e-mail addresses of two authors, Geum Jin Kim and Sang Hee Shim, were changed as their institutional email addresses in the newly submitted manuscript.

- In page 1, line 24: The phrase “menopausal symptoms” was revised as “estrogen reduction”.

- In page 1, line 26: The phrase “a uterotrophic” was corrected as “an uterotrophic”.

- In page 1, line 27: The description “and aerial parts” was changed to “part”.

- In page 1, lines 32: The phrase “enhance various menopausal symptoms” was changed as “have estrogenic effects”.

- In page 2, line 83: The phrase “menopausal symptoms” was revised as “estrogen reduction”.

- In page 3, line 100: Effect of ICI 182,780 on MCF-7 cell proliferation was added to Figure 2.

- In page 3, line 102: The unit of ICI 182,780 was changed to micromolar.

- In page 4, lines 112-114: A short description on the concentration independent estrogenic activity by SA-bg-E50 was added.

- In page 4, line 128: A description “relieve menopausal symptoms” was modified as “have estrogenic effects”

- In page 5, lines 142-143: Unclear description was revised and now it reads as follows: The results indicate that SA-bg-E50 hypertrophies the uterus owing to its estrogenic activity (Figure 3).

- In page 5, line 145: The style of compound number was revised as bold style. 

- In page 6, line 197: The description of “Belowground and Aerial Parts” was changed to “Belowground Part”.

- In page 6, line 199-200: The phrase “belowground and aerial parts” was revised as “belowground part”.

- In page 6, lines 200, 203, 204, 207 and 209: The format of compound number was revised as bold style.

- In page 6, lines 201 and 205: The units of test samples were corrected as “μM”.

- In page 6, lines 204-208: The descriptions on estrogenic activities of compounds 1 and 2 were revised.

- In page 6, line 211: In the description of figure 6, “belowground and aerial parts” was corrected as “belowground part”.

- In page 6, lines 219-220: A sentence describing the additional experiments on ERβ was add.

- In page 7, lines 224-225: The descriptive sentence on the phosphorylation on ERβ by compound 1 and 17β-estradiol treatment was added.

- In page 7, lines 227-234: Bar chart for the phosphorylation of ERβ by compound 1 and 17β-estradiol treatment was added. The description of figure 6 was revised as the figure changed.

- In page 7, line 238: The sentence “Thus, compound 1 is expected to relieve various menopausal symptoms.” was removed.

- In page 7, line 245: Two references supporting side effects of HRT were added.

- In page 7, lines 247-251: A paragraph describing the breast cancer development by estrogen-like compounds along with references were added.

- In page 8, line 253: A cell proliferation data of tamoxifen was added and the test concentrations of compounds 1-5 were corrected.

- In page 8, lines 254-264: In the description of figure 7, the descriptions were revised as the figure modified.

- In page 8, line 270: The phrase “relieve various menopausal symptoms” was revised as “have estrogenic effects”.

- In pages 9-10, lines 296-323: Descriptions on the isolation of compounds 1-5 were revised.

- In page 10, lines 335-339: Description on Queens one tablet was added as requested.

- In page 10, lines 344 and 353: “S.D.” was corrected as “SEM”.

- In page 10, lines 348, 356: The reference number changed from 43 to 48.

- In page 10, line 358-359: The additional experiment related with ERβ was described additionally.

- In page 10, lines 366-368: Experimental condition was corrected to prevent unnecessary confusion.

- In page 11, line 388: The reference number 44 was revised to 48.

- In page 11, line 389: The phrase “a uterotrophic” was changed to “an uterotrophic”.

- In page 11, line 402: The detail on graphic software was added.

- In page 11, lines 407 and 410: The term “menopausal symptoms” was corrected to be “estrogen reduction”.

- In page 11, lines 416-418: The final conclusion on the potential of compound 1 was revised.

- In pages 13-14, lines 506-514: Five references were added in the reference list.

- In page 14, lines 515-518: The reference number 43 and 44 was changed to 48 and 49.

List of Changes in the Supplementary Materials

- The e-mail addresses of two authors, Geum Jin Kim and Sang Hee Shim, were changed as their institutional email addresses.

Round 2

Reviewer 1 Report

The authors revised the manuscript precisely.

This version is worthy of publication.

Reviewer 2 Report

Accept in the present form.

Reviewer 3 Report

All of my comments are addressed.